# Extensive CArdioVAscular Characterization and Follow-Up of Patients Receiving Immune Checkpoint Inhibitors: A Prospective Multicenter Study

**DOI:** 10.3390/ph16040625

**Published:** 2023-04-20

**Authors:** Danielle Delombaerde, Johan De Sutter, Lieselot Croes, Delphine Vervloet, Veronique Moerman, Nico Van de Veire, Anne-Marie Willems, Kristien Wouters, Marc Peeters, Hans Prenen, Christof Vulsteke

**Affiliations:** 1Integrated Cancer Center Ghent, Department of Medical Oncology, AZ Maria Middelares, 9000 Ghent, Belgium; 2Center for Oncological Research (CORE), Integrated Personalized and Precision Oncology Network (IPPON), University of Antwerp, 2610 Wilrijk, Belgium; 3Department of Cardiology, AZ Maria Middelares, 9000 Ghent, Belgium; 4Faculty of Medicine and Health Sciences, Ghent University, 9000 Ghent, Belgium; 5Department of Cardiology, Free University Brussels, 1000 Brussels, Belgium; 6Antwerp University Hospital, Clinical Trial Center (CTC), CRC Antwerp, 2650 Edegem, Belgium; 7Multidisciplinary Oncologic Center Antwerp (MOCA), Antwerp University Hospital, 2650 Edegem, Belgium

**Keywords:** immune checkpoint inhibitor, immune-related adverse event, cardiotoxicity, cardiac troponin, myocarditis, biomarker, cardio-oncology, subclinical cardiotoxicity, diastolic function

## Abstract

Background: The increasing use of immune checkpoint inhibitors (ICIs) in the treatment of both advanced and early stages of various malignancies has resulted in a substantial increase in the incidence of cardiovascular (CV) immune-related adverse events (irAEs). The current follow-up guidelines are based on anecdotal evidence and expert opinions, due to a lack of solid data and prospective studies. As many questions remain unanswered, cardiac monitoring, in patients receiving ICIs, is not always implemented by oncologists. Hence, an urgent need to investigate the possible short- and long-term CV effects of ICIs, as ICI approval is continuing to expand to the (neo)adjuvant setting. Methods: We have initiated a prospective, multicenter study, i.e., the CAVACI trial, in which a minimum of 276 patients with a solid tumor, eligible for ICI treatment, will be enrolled. The study consists of routine investigations of blood parameters (troponin and N-terminal pro-B-type natriuretic peptide (NT-proBNP) levels, in particular) and a thorough CV follow-up (electrocardiograms, transthoracic echocardiograms, and coronary calcium scoring) at fixed time points for a total period of two years. The primary endpoint is the cumulative incidence of troponin elevation in the first three months of ICI treatment, compared to baseline levels. Furthermore, secondary endpoints include incidence above the upper limit of normal of both troponin and NT-proBNP levels, evolution in troponin and NT-proBNP levels, the incidence of CV abnormalities/major adverse cardiac events, evaluation of associations between patient characteristics/biochemical parameters and CV events, transthoracic echocardiography parameters, electrocardiography parameters, and progression of coronary atherosclerosis. Recruitment of patients started in January 2022. Enrolment is ongoing in AZ Maria Middelares, Antwerp University Hospital, AZ Sint-Vincentius Deinze, and AZ Sint-Elisabeth Zottegem. Trial registration: ClinicalTrials.gov Identifier: NCT05699915, registered 26 January 2023.

## 1. Introduction

Immune checkpoint inhibitors (ICIs) have significantly altered the field of oncology during the last decade by improving clinical outcomes in multiple cancer types. Currently, the U.S. Food and Drug Administration (FDA) has approved eight ICIs for over 50 indications. Nevertheless, the increasing use of ICIs in the treatment of both advanced and early stages of various malignancies has resulted in a substantial increase in immune-related adverse events (irAEs). The most frequently observed irAEs occur in the skin, endocrine system, and gastrointestinal tract. The majority of irAEs can be managed effectively if found and treated early. Other irAEs, i.e., myocarditis, pneumonitis, encephalitis, and hepatitis, are less frequently seen, though can be potentially fatal.

More specifically, immune-related cardiovascular (CV) events (myocarditis, pericardial disease, peri myocarditis, vasculitis, left ventricular dysfunction, acute coronary syndromes, and arrhythmias and cardiac conduction abnormalities) have gained significant interest over the last few years due to their high mortality rate e.g., ICI-induced myocarditis (39.7 to 50%) [1,2,3,4,5,6,7,8,9,10,11]. Moreover, there is increasing evidence that cardiac irAEs have been missed in a non-negligible number of patients during the initial years of ICI approvement, due to the wide varieties of clinical presentation [12]. Furthermore, the lack of routine monitoring for cardiac events and the inclusion of a highly-selected and healthier patient population in immunotherapy trials has most likely contributed to the under-reporting of CV irAEs. As ICIs evolve to include high-risk patients with pre-existing CV risk factors and disease, as well as patients receiving ICIs in the adjuvant setting, the risk of ICI-induced cardiotoxicities is highly relevant.

A recent systematic review and meta-analysis of 125 trials, which included 20,218 patients treated with either anti-programmed cell death protein-1 (PD-1) or anti-programmed cell death ligand-1 (PD-L1), revealed 82 treatment-related deaths. Importantly, of these 82 deaths, there was a 9.8% incidence of CV death, including myocardial infarction and acute coronary syndromes. The important role of cytotoxic T-lymphocyte-associated antigen-4 (CTLA-4) has already been suggested 25 years ago in a CTLA-4 deficient mouse strain with the development of lymphoproliferative disease with multiorgan lymphocytic infiltration and tissue destruction, with particularly severe myocarditis and pancreatitis, and death after 3–4 weeks of age. Recently, in patients treated with ICIs for melanoma, a significantly increased uptake of 2-[18F]fluorodeoxyglucose in the large arteries was observed which could indicate vascular inflammation. Vascular inflammation is a well-known contributor to atherosclerosis and heart failure with preserved ejection fraction. Accordingly, recent clinical data has shown a three-fold higher risk of CV events after ICI initiation, as well as a three-fold higher rate of progression of total aortic plaque volume, thus suggesting that CV irAEs could be mediated by an accelerated progression of atherosclerosis [8,13,14,15,16,17,18,19,20,21,22,23,24,25,26].

Several international guidelines recommend preventive/screening measures for CV irAEs as a safety margin. More specifically, the current guidelines (NCCN, ASCO, ESC, and SITC) recommend an electrocardiogram (ECG) and a CV risk assessment in all patients prior to ICI initiation [6,9,27,28,29,30,31,32,33]. Baseline echocardiography is only recommended in high-risk patients (dual ICI therapy, combination of ICI therapy with other cardiotoxic therapies, prior cancer therapy-related cardiac dysfunction, or prior CV disease), however, it can be considered in all patients (level C evidence) [33]. In addition, they also recommend the baseline determination of cardiac markers in all patients, i.e., troponin and brain natriuretic peptide (BNP), or its prohormone N-terminal pro-B-type natriuretic peptide (NT-proBNP). As cardiac irAEs often tend to occur during the first three months of treatment, most guidelines also recommend serial troponin measurements prior to each cycle during the first weeks of treatment, especially in patients with an abnormal baseline value [1,6,9,29,31,34,35,36].

Cardiac markers such as high-sensitivity troponin I/T (hs-TnI/hs-TnT), BNP, and NT-proBNP are crucial in the diagnosis of CV diseases. For example, troponin is a well-known cardiac biomarker that has become indispensable in the diagnosis of acute coronary syndromes. However, its role in cardio-oncology, more specifically in the active surveillance of ICI-mediated CV events, is not clear. Hence, the current guidelines are based on anecdotal evidence and expert opinions, due to the lack of solid data and prospective studies [33,37]. Therefore, cardiac monitoring, in patients receiving ICIs, is not always implemented by oncologists, as many questions remain unanswered:How should hs-TnI/hs-TnT levels be graded? The Common Terminology Criteria for Adverse Events (CTCAE, Version 5) is widely accepted as the standard classification and severity grading scale for adverse events in cancer therapy, clinical trials, and other oncology settings [38]. However, the current grading system makes it challenging to accurately document and report the severity and incidence of increased cardiac troponins, leading to ICI-induced CV toxicity. Furthermore, BNP and NT-proBNP are not even listed in the current CTCAE criteria.What are the cut-off values for cardiac markers for clinically meaningful changes in ICI-treated patients? When do further investigations need to be performed? (The upper limit of normal (ULN) is determined on a healthy patient population).What is the appropriate time interval between the testing of cardiac markers?What is the appropriate monitoring duration of cardiac markers? Three months or longer?

Our narrative review summarized previously conducted research regarding troponin levels in patients receiving ICI [39]. As prospective data are lacking, we initiated a multicenter, prospective trial (NCT05699915) in which patients with a solid tumor eligible for ICI treatment, i.e., anti-CTLA-4, anti-PD-1, and/or anti-PD-L1, will be enrolled. Patients receiving concurrent systemic antineoplastic treatment will be excluded in order to have a homogeneous treatment population. The patients included in this study will receive a thorough CV work-up at baseline and at regular time points during treatment for a total period of two years. Our aim is to study cardiac markers in patients receiving ICI therapy and to explore the association between CV abnormalities and major adverse cardiac events (MACEs).

## 2. Methods and Analysis

### 2.1. Study Flow

The CAVACI trial, i.e., extensive CArdioVAscular characterization and follow-up of patients receiving immune checkpoint inhibitors, is a prospective, multicenter study. The study was approved by the central ethics committee of the Antwerp University Hospital (2021–1908, 2022–1908) and follows the standards of the Declaration of Helsinki and the ethical standards of the responsible committee on human experimentation. The study consists of routine investigations of biochemical parameters and a thorough CV follow-up at fixed time points. Patients included in the study will be followed for a total period of two years (Figure 1). Patients are currently being recruited via the multidisciplinary oncology board of each hospital. As of January 2022, 57 patients were enrolled.

### 2.2. Participants

#### 2.2.1. Inclusion Criteria

All patients must fulfill the following inclusion criteria:At least 18 years of age, at the time of giving informed consent.Able to provide informed consent.Have a solid tumor and will receive one of the following FDA-approved therapies, i.e., anti-PD-1, anti-PD-L1, and/or anti-CTLA-4 therapy in mono- or combination therapy.Have to be literate in Dutch or English.

#### 2.2.2. Exclusion Criteria

Prior treatment with immunotherapy (ICIs, T-cell transfer therapy, cancer treatment vaccines, or immune system modulators).Receive a regimen where ICIs will be administered together with other systemic anti-cancer agents (chemotherapy, tyrosine kinase inhibitors (TKI), etc.).History of human immunodeficiency virus infection.History of hepatitis B (defined as hepatitis B surface antigen [HBsAg] reactive) or known hepatitis C virus (HCV) (defined as detectable HCV RNA via qualitative nucleic acid testing) infection.Diagnosis of immunodeficiency or receiving chronic systemic steroid therapy (in daily doses exceeding 10 mg of prednisone equivalent).

### 2.3. Schedule of Examinations

#### 2.3.1. Baseline Patient/Disease Characteristics

Upon enrolment, the following data will be collected:Informed consent.Demographics (Eastern Cooperative Oncology Group performance status, age at start of treatment, sex, body mass index).Medical history: CV risk, COVID-19, auto-immune diseases, and other medical conditions.Current oncological disease (clinical and pathological TNM classification, current disease status, prior treatment, and molecular screening profile).Prior cancer history.Prior/concomitant medication review, especially antibiotic and oral steroid use.Other relevant parameters.

#### 2.3.2. Electrocardiogram (ECG)

A standard resting 12-lead ECG will be obtained (Appendix A) using an ECG machine (GE Healthcare, Acertys, Horten, Norway) and will be analyzed for intervals, wave vectors, and morphology, ST-segment changes and corrected QT interval (QTc) using Bazett’s Formula.

#### 2.3.3. Transthoracic Echocardiogram (TTE)

A TTE will be performed in all patients at baseline, 3, 6, 12, and 24 months (Appendix A). A comprehensive evaluation of the systolic and diastolic function, ventricular and atrial geometry, will be performed following the guidelines by the American Society of Echocardiography on a Vivid E95 ultrasound system (GE Healthcare, Horten, Norway) [40]. Special attention will be given to acquiring a 3D measurement of the left ventricular ejection fraction (LVEF) and performing deformation imaging of the left ventricle global longitudinal strain (GLS). The right ventricular function will be evaluated by tricuspid annular plane systolic excursion (TAPSE) and peak systolic velocity S’ derived from color coded tissue Doppler imaging (TDI). Diastolic dysfunction will be based on an average E/e′ ratio > 15 and a LA area > 30 cm^2^.

#### 2.3.4. Computed Tomography (CT) Scan for Calcium Scoring

A CT-scan will be performed at baseline, 12 months, and 24 months in order to detect calcium deposits in the coronary arteries of the heart (Appendix A). A higher coronary calcium score suggests a higher chance of significant narrowing in the coronary arteries.

#### 2.3.5. Blood Sampling

Blood samples of all participants will be collected at baseline (immune baseline, four tubes i.e., serum, EDTA, fluoride, and one EDTA tube on ice) and before each ICI cycle (immune follow-up, three tubes i.e., serum, EDTA, and fluoride) (Appendix A). One additional blood sample (serum) will be taken at baseline, 3, 6, 12, and 24 months at the same time of routine laboratory testing.

### 2.4. Endpoints

The objective of this study is to investigate troponin and NT-proBNP values in patients receiving ICIs and their association with ICI-induced CV abnormalities and MACEs. Furthermore, we will study the calcium score, and systolic and diastolic function in this patient population. We will evaluate the associations between patient/disease characteristics/transthoracic echocardiography parameters/electrocardiography parameters and troponin/NT-proBNP levels.

#### 2.4.1. Primary Endpoint

The incidence of an elevated hs-TnT above the ULN if the baseline value was normal; or 1.5 ≥ times baseline if the baseline value was above the ULN within the first three months of treatment. The maximum measured value will be taken into account *.

#### 2.4.2. Secondary Endpoints

The incidence of hs-TnT/NT-proBNP elevations at 6, 12, and 24 months.The incidence of hs-TnT/NT-proBNP above the ULN at baseline, 3, 6, 12, and 24 months *.Evolution of hs-TnT/NT-proBNP in 24 months compared to baseline.Evolution of transthoracic echocardiography parameters at baseline, 3, 6, 12, and 24 months *.Evolution of electrocardiography parameters at baseline, 3, 6, 12, and 24 months.*Association between the evolution of troponin/NT-proBNP and transthoracic echocardiography and electrocardiography parameters at baseline, 3, 6, 12, and 24 months *.Cumulative incidence of CV abnormalities at 3, 6, 12, and 24 months based on the CARDIOTOX classification system of Sendón et al. (Table 1), with the inclusion of pericardial effusion and new arrhythmias [41]:

**Table 1 pharmaceuticals-16-00625-t001:** Cardiovascular abnormalities will be based on the CARDIOTOX classification system of Sendón et al. [41]. We will also document pericardial effusion and new arrhythmias as cardiovascular abnormalities.

Normal	Patients with Normal Biomarkers and LV Function Parameters
Mild	Asymptomatic patients with LVEF ≥ 50% with elevated biomarkers Asymptomatic patients with LVEF ≥ 50% with at least one additional abnormal echo parameter: (1) Increased LVESV (2) LA area > 30 cm^2^ (3) 10% decrease of LVEF to an LVEF < 53% (4) Average E/e’ > 14 (5) GLS > −18% (6) 15% relative reduction of GLS from baseline
Moderate	Asymptomatic patients with LVEF ≥ 40% and <50% with or without biomarker increase or other LV function abnormalities
Severe	Patients with asymptomatic LVEF < 40% Clinical HF: - HFrEF: HF symptoms/signs and LVEF < 40% - HFmrEF: symptoms/signs of HF with elevated NT-proBNP, LVEF 40–49%, and at least one additional criteria (enlarged LA, LV hypertrophy, or other relevant diastolic function parameters) - HFpEF: in presence of symptoms/signs of HF, elevated NT-proBNP, LVEF ≥ 50%, and at least one additional criteria (enlarged LA, LV hypertrophy, or other diastolic dysfunction parameters)

Abbreviations: GLS, global longitudinal strain; HF, heart failure; HFmrEF, HF with mildly-reduced ejection fraction; HFpEF, HF with preserved ejection fraction; HFrEF, HF with reduced ejection fraction; LA, left atrial; LV, left ventricular; LVEF, left ventricular ejection fraction; LVESV, left ventricular end-systolic volume; NT-proBNP, N-terminal pro-B-type natriuretic peptide.

Association between the evolution of troponin/NT-proBNP and CV abnormalities.Cumulative incidence of MACEs at 3, 6, 12, and 24 months. MACEs were defined as the composite outcomes of nonfatal stroke, nonfatal myocardial infarction, hospital admission for heart failure (HF), cardiac revascularization, and CV death.Overall survival *.Association between the evolution of troponin/NT-proBNP and MACEs.The difference in the evolution of hs-TnT/NT-proBNP/transthoracic echocardiography and electrocardiography parameters between combination therapy and monotherapy.Association between patient characteristics and troponin.Association between patient characteristics and NT-proBNP.Agreement between hs-TnT and hs-TnI levels at baseline, 3, 6, 12, and 24 months *.The proportion of severe immune-related non-CV toxicities (grades 3–5).Association between the evolution of troponin/NT-proBNP and severe immune-related non-CV toxicities (grade 3–5).Association between the evolution of troponin/NT-proBNP and overall survival.Association between the evolution of troponin and diastolic function (based on the recommendations of Nagueh et al. [42]).Association between the evolution of troponin and calcium score.

* These endpoints will also be evaluated in a preliminary analysis once 50 patients have reached their 3-month cardiac follow-up visit. The sample size was not adjusted for interim analysis as the release of preliminary results is not data-driven and does not affect the error structure of the trial. Study-level conclusions will only be made once the final results of the trial have been analyzed.

### 2.5. Sample Size

Our primary endpoint is the cumulative incidence of troponin elevation in the first three months of ICI treatment, compared to baseline levels. In order to obtain a sufficiently precise estimate of this incidence, the sample size was calculated based on the width of the 95% confidence interval.

As of January 2021, we added troponin to our standard-of-care blood analyses in ICI-treated patients. Approximately 10% of our patients had troponin elevation according to the definition in Section 2.4.1. Therefore, we assumed a 3-month cumulative incidence of 10% in our sample size calculation. To estimate this incidence with a precision of 7.5%, we need to include 249 patients. Precision is hereby defined as the width of the 95% confidence interval.

The dropout rate in the first three months is anticipated to be around 10%, mainly due to deaths. Therefore, death will be accounted for as a competing risk in the primary analysis. Simulations using Weibull distribution for time to death and time to troponin elevation showed that adding 10% in the calculated sample size still leads to a precision of less than 7.5% in the estimation of the cumulative incidence with death as a competing risk. Therefore, 276 patients will be included in the study.

### 2.6. Data Analysis

All patient data will be collected via REDCap (a Research Electronic Data Capture), a secure HIPAA (Health Information Portability and Accountability Act)-compliant Web-based application, using a standardized data collection form. Data needed for analyses will be extracted from REDCap [43,44]. For categorical variables, frequencies and percentages will be reported. Where values are missing, percentages will be calculated for the available cases, and the denominator will be mentioned. Continuous variables will be summarized as mean with standard deviation and range.

For the primary endpoint, the cumulative incidence of troponin elevation will be calculated with death as a competing risk. Cumulative incidences and corresponding 95% confidence intervals will be reported at 3, 6, 12, and 24 months and a cumulative incidence plot will be used to visualize the results.

Similarly, for the secondary endpoints elevation of NT-proBNP, CV abnormalities, and MACE, cumulative incidences and 95% confidence intervals will be calculated, considering death as a competing event.

Evolution of troponin, NT-proBNP, transthoracic echocardiographic, and electrocardiographic parameters over time will be first studied in a linear mixed effects model with a random intercept per subject to account for the correlation of measurements coming from the same individual. This model will be extended with patient and treatment characteristics and their interaction with time, to evaluate their impact on the evolution of these parameters.

The association between the longitudinal evolution of troponin levels and time-to-event endpoints (CV abnormality, MACE) will be studied in a joint model combining a linear mixed model for troponin and a sub-distributional proportional hazards model for the time-to-event taking into account death as a competing event for CV abnormality and MACE. Latent random variables and common covariates are used to link the sub-models for the longitudinal measurements with the competing risk failure time data. In this way, the association between the evolution of the longitudinal measurements and time to event can be evaluated. Furthermore, by modelling the time to event, the analysis of the longitudinal measurements is adjusted for non-ignorable missing data due to informative dropout caused by deaths.

Agreement between hs-TnT and hs-TnI (measured in all patients at baseline, 3, 6, 12, and 24 months) will be assessed in Bland–Altman curves and intraclass correlation coefficient (ICC) based on a two-way mixed effects model. The ICC and 95% confidence interval will be reported. Statistical analyses will be performed using R Software version 4.1.2 or higher.

As our clinical trial is an exploratory trial, secondary outcomes will be treated as exploratory results. Therefore, any definite finding for secondary outcomes will require further confirmatory studies to support possible findings. Hence, it is not needed to adjust for multiple testing for multiple secondary outcomes [45].

## 3. Discussion

Since 2020, CV follow-up has become a “hot topic” in ICI-treated patients. Shortly after our study was initiated, other research groups started publishing their results on similar CV screening programs in patients receiving ICIs.

To determine the role of GLS, global circumferential strain (GCS), and global radial strain (GRS) in ICI-treated patients, Quinaglia et al. conducted a retrospective study in which 75 patients with ICI myocarditis were compared with 50 ICI-treated patients without myocarditis [46]. They found that GLS, GCS, and GRS can predict CV events with better accuracy than LVEF, hs-TnT, and age, in patients who were diagnosed with ICI-induced myocarditis. However, only limited paired data were available for the analysis. Furthermore, pre-ICI TTEs were not investigated and TTEs during treatment were performed at different time points.

Kurzhals et al. retrospectively analyzed pre-treatment hs-TnT levels in 47 patients who subsequently received ICI therapy for locally advanced and/or metastatic melanoma [47]. One patient, who had normal pre-treatment hs-TnT, developed myocarditis. 28% of the patients had elevated levels at baseline, however, none developed ICI-induced myocarditis. There was no association between hs-TnT and overall survival. However, patients were only monitored for a short period of time.

Waissengein et al. performed a retrospective analysis in 135 patients treated with first-line pembrolizumab to evaluate whether baseline and longitudinal changes in hs-TnI could serve as a predictor for the development of MACEs and survival [48]. They found that abnormally elevated baseline hs-TnI levels (>50 ng/L) and levels prior to the second dose served as significant independent predictors for MACEs, with over an eightfold increase in relative risk. In addition, elevated baseline levels had a predictive role in all-cause mortality. However, troponin levels were only evaluated prior to the start and at cycle two. Echocardiography was only performed in 53 patients at baseline and 27 patients in follow-up. Therefore, the role of hs-TnI in the development of LV dysfunction could not be assessed. Moreover, some patients (27%) also received pembrolizumab in combination with other types of chemotherapy, which can also influence the effect on MACE and survival.

In a recent retrospective cohort study, Tamura et al. found that increased hs-TnI levels were associated with an early worsening in both global and regional longitudinal strain in patients receiving ICI therapy [49]. 18 patients had elevated hs-TnI levels of which six were diagnosed with myocarditis. One of the limitations is that the study was conducted in a single center in Japan. Three of the 129 included patients had melanoma, whereas melanoma is more common in the Caucasian patient population.

The same research group also studied cardiac complications in patients receiving long-term ICI therapy (>6 months) [50]. They conducted a single-center pilot study in which they retrospectively analyzed 55 patients who received single-agent ICI therapy together with routine cardiology follow-up visits. None of the patients met the primary endpoint, i.e., discontinuation of ICI due to cardiac events. Four of the patients had elevated hs-TnI levels, seven had a decline in GLS values, and two had elevated BNP levels. None of the patients developed new cardiac abnormalities after two years of treatment.

Furukawa and Tamura et al. also initiated a prospective, single-center screening program for myocarditis in ICI-treated patients. A total of 126 patients were enrolled, who received a thorough CV follow-up prior to therapy, after 7, 14, 21, and 60 days (vitals, biomarkers, ECGs, chest CTs, echocardiographs) [50]. A total of 13 out of 18 patients who developed hs-TnI elevations, had signs of clinically suspected myocarditis. No associations were found with known risk factors of CV diseases. However, they did notice that creatine kinase (CK) was elevated in the four patients that had moderate to severe myocarditis, based on the ESC position statement [51]. Thus, suggesting that CK could be a useful pre-onset biomarker for ICI-related myocarditis. Despite the prospective nature of the study, all outcomes were retrospectively analyzed.

Vasbinder et al. also found that an increase in CK was associated with the development of myocarditis and all-cause mortality. They conducted a retrospective observational cohort study in 2606 ICI-treated patients, in which they discovered that the long-term survival was similar in patients with ICI-induced myocarditis and patients without ICI myocarditis [52]. However, hs-TnT levels were not systematically measured in patients without ICI myocarditis (n = 2579), and thus no comparisons were possible.

The Javelin Renal 101 phase III trial (NCT02684006) was the first randomized trial to include prospective serial cardiac imaging (echocardiography or multigated acquisition (MUGA) scan to measure changes in LVEF) and serum cardiac biomarkers (hs-TnT, hs-TnI, BNP, NT-proBNP, and CK–myoglobin binding) [53]. Rini et al. discovered that patients receiving avelumab along with axitinib who had high baseline hs-TnT values were at higher risk of MACEs than the patients with low values. This association was not observed in the sunitinib group. Only LVEF was investigated on cardiac imaging. An ECG was only taken at baseline, however, results were not reported nor investigated. Furthermore, as both avelumab and axitinib can be potentially cardiotoxic, no distinction could be made in the drug causing MACE. Another important note is that biomarker testing as well as cardiac imaging were not standardized between the different sites.

Xu et al. recruited 55 patients eligible for ICI therapy in order to evaluate subclinical cardiac dysfunction by using 2D speckle tracking imaging and 3D echocardiography (Philips IE33 ultrasound) [54]. The results were analyzed at 220 days of follow-up. They found that the left ventricular global longitudinal peak systolic strain (LVGLS), right ventricular global longitudinal systolic strain, and TAPSE significantly deteriorated after ICI treatment. LVGLS was found to be more sensitive to detecting subclinical cardiac dysfunction than CV toxicity events (HF, reduction in LVEF, and increase in hs-TnI). However, it remains unclear at what exact time post-immunotherapy patient follow-up happened. In addition, many of the patients who received ICI therapy in combination with chemotherapy and patients with an important CV history were excluded. Furthermore, the study had a small sample size and only included Chinese patients.

Systematic screening was also performed in a prospective, single-center study by Faubry et al. which included 99 patients [55]. The main objective was to determine the incidence of myocarditis and other CV irAEs in ICI-treated patients with stage IIIB-IV lung cancer. They all received baseline measurements of hs-TnI and NT-proBNP together with an ECG and echocardiogram. The majority of the patients (67%) were being treated with a combination of ICIs and chemotherapy, which both can result in cardiotoxicity. A cumulative incidence rate of 3% was found for ICI -induced myocarditis during a 6-month follow-up. In addition, myocarditis also had later onset than previously reported [5,56]. Although cardiac biomarkers and ECGs were taken prior to each ICI cycle for six months, no routine echocardiography was performed.

Patients with non–small–cell lung cancer, receiving ICI monotherapy, were investigated by Isawa et al. in a prospective observational study [57]. One hundred and twenty-nine patients were enrolled and received serial cardiac monitoring, i.e., hs-TnT, BNP, and an ECG at baseline and every four to six weeks thereafter. A TTE was only performed at baseline. Patients with preceding grade 1 CV irAES (asymptomatic CV irAEs) had a significantly higher risk of developing ≥ grade 2 CV irAEs. Patients with prior acute coronary, prior heart failure hospitalization, or disease control were significantly associated with grade ≥ 1 CV irAEs. However, quantitative troponin data were not obtained. Furthermore, as monitoring was performed at every other treatment cycle some events might have been missed.

Nishikawa et al. conducted a prospective study on 100 patients eligible for ICI treatment [58]. An ECG was taken every month along with hs-TnI and NT-proBNP measurements. A 3D echocardiography (IE 33 imaging device, Philips Healthcare, Amsterdam, The Netherlands) was taken before treatment, at three months, and six months in order to determine the LVEF (modified Simpson method) and GLS. Myocardial damage was defined as an increase in hs-TnI levels by >26 pg/mL (Manufacturer’s cut-off value) and/or a decrease in LVEF by >10% to <53% on echocardiography. Ten patients (10%) developed myocardial damage within six months of treatment initiation. An increase in hs-TnI levels was noted in all ten patients, except one. However, this patient did have a decrease in LVEF from 57% to 41% along with an increase in NT-proBNP levels. GLS decreased by >15% in five of the patients. The authors noted that serial cardiac troponin I measurements could help detect early-phase myocardial damage by ICIs. It is important to note that 21 out of 100 patients received ICI therapy in combination with other systemic anti-cancer agents. Four of the ten patients that had myocardial damage received an ICI in combination with another systemic antineoplastic agent. Thus, the true cause of myocardial damage in these patients remains unknown. Furthermore, the patient’s risk factors for myocardial damage as well as prognosis were not further investigated in this study.

In conclusion, data regarding diagnosis, screening, and treatment recommendations on CV irAEs are insufficient. Guidelines that have been developed, however, are mostly based on expert opinions, retrospective studies, and case reports. Lately, more research has been conducted, however, should not only focus on the prediction and detection of ICI-induced myocarditis, as many other CV irAEs exist and should be taken into account. The results of this study will hopefully provide a better insight into cardiac biomarkers in ICI-treated patients and additional information in order to refine the current evidence-based screening recommendations and optimize risk stratification. Data from the CAVACI trial will be submitted to a peer-reviewed journal for publication.

## Figures and Tables

**Figure 1 pharmaceuticals-16-00625-f001:**
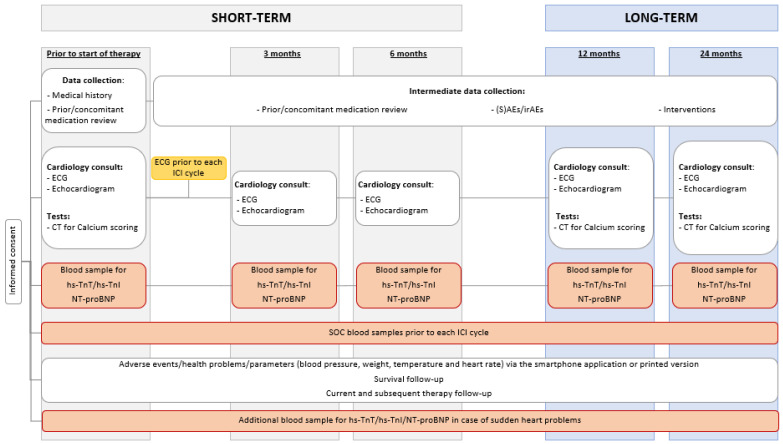
Study flow. Abbreviations: CT, computed tomography; ECG, electrocardiogram; Hs-TnI, high-sensitivity troponin I; hs-TnT, high-sensitivity troponin T; ICI, immune checkpoint inhibitor; irAE, immune-related adverse event; NT-proBNP, N-terminal pro-B-type natriuretic peptide; sAE, severe adverse event; SOC, standard of care.

## Data Availability

Not applicable.

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
