# Peer review of "Extensive CArdioVAscular Characterization and Follow-Up of Patients Receiving Immune Checkpoint Inhibitors: A Prospective Multicenter Study"

_pharmaceuticals, 2023, doi:10.3390/ph16040625_

Round 1

Reviewer 1 Report

In this paper, Delombaerde et al present the design of a study aiming to investigate the possible cardiovascular immune-related adverse events (irAEs) resulting from treatment with immune checkpoint inhibitors (ICIs). Indeed, ICIs have been shown to provoke other forms of irAEs, particularly gastrointestinal irAEs. Therefore, there is a rationale to investigate the possible cardiovascular side effects of these drugs, in the absence of reliable data in the literature.

Overall, the study is of interest, and is well thought out. Its results will certainly be of use to the scientific community by providing robust information from systematic examination of patients taking ICIs. Since the paper presents the study design, the issues I have mentioned are merely for the sake of intellectual debate, since I understand that it is probably too late now to change any key design components of the study, which is already under way.

I wonder whether the primary endpoint of troponin elevation is the most meaningful endpoint. Indeed, troponin elevation might not necessarily capture heart failure, for example. If the authors were keen to use biomarkers, perhaps a composite endpoint, comprising elevation of any one of a set of pertinent cardiovascular biomarkers including BNP, NT-proBNP, MR-proANP or D-dimers, as well as troponin, might have been more useful. However, I do note that these are included (except perhaps D-dimers and Mr-proANP) in the secondary endpoints, so any changes will probably be captured nonetheless.

Another point is inflammation: do the authors plan to measure routine markers of inflammation/inflammatory response? I didn’t see this in the text (maybe I missed it?) and it would be helpful if the authors could clarify.

For a study that has not yet completed inclusions, I think the discussion is too long and could easily be shortened without prejudice.

Finally, here are some suggestions for minor grammatical corrections:

-          Page 1, line 29, “at” fixed timepoints, not “on”; same remark again on page 3, line 121.

-          Page 10, line 370 – the word “distinguishment” does not exist. Please correct to “distinction”.

-          Page 10, line 387 – similarly, the word “monocentric” also does not exist. Please correct to “single-centre”.

-          Page 10, line 392 – should read “…treated with a combination of”…

-          Page 10, line 392 – please change “effect” (noun) to “affect” (verb)

-          Page 10, lines 394-395 – please remove the “a” at the beginning of line 395, so that the text reads “…myocarditis also had later onset…”

-          Page 10, line 396, remove the apostrophe and the s at the end of the word “echocardiography” and put the verb in the singular, so that it reads “…no routine echocardiography was incorporate into the screening programme”.

-          Page 10, line 399 – it is generally not acceptable practice to start a sentence with numbers. Please spell out in words.

Reviewer 2 Report

I reviewed with interest the manuscript by Delombaerde et al. "Extensive CArdioVAscular characterization and follow-up of patients receiving immune Checkpoint Inhibitors: a prospective multicenter study". In this article, the authors presented the protocol of a prospective study conducted since January 2022. The protocol has been sufficiently developed, including registration on the ClinicalTrials.gov website (NCT05699915). However, the authors considered it necessary to publish it also as an article in a journal, so I consider it necessary to make comments and ask questions to the authors of the article.

1. How relevant is the assessment of coronary artery calcification for cancer patients? The authors rightly point out that atherosclerosis can develop over decades, and the life span of patients in the study is clearly shorter. In addition, in the treatment of ISI, patients develop myocarditis as cardiotoxic manifestations (it is precisely for its diagnosis that the study of biomarkers - troponin, natriuretic peptides - and ECHOCG indicators) is aimed. It is unlikely that such myocarditis will lead to the rapid development of atherosclerosis. Perhaps the authors have such data, then they should be given in the rationale for the study, or in its discussion.

2. Among the ECHOCG parameters, the authors plan to study primarily left ventricular parameters (including 3D measurement of left ventricular ejection, GLS and left ventricular diastolic function). At the same time, the assessment of right ventricular function is more modest (TAPSE and peak systolic velocity S'), although there are more informative indicators (3d-ECHOCG, RV diastolic function, global deformation along the longitudinal axis). Since changes in the right ventricle may develop earlier than changes in the left ventricle (1,2), they may well be used to detect subclinical myocardial changes during ISI therapy (3).

3. The number of secondary endpoints in the proposed study raises concerns. With such a number of them, there will inevitably be the influence of multiple comparisons, which will need to be taken into account in statistical analysis. However, in the study design, the authors do not describe such a statistical procedure.

4. Some of the sources in the list of references are given incorrectly. For example: D'souza M, Nielsen D, Svane IM, Iversen K, Rasmussen PV, Madelaire C, et al. The risk of cardiac events in patients receiving immune checkpoint inhibitors: a nationwide Danish study n.d. (??) doi:10.1093/eurheartj/ehaa959. or Dolladille C, Akroun J, Morice P-M, Dompmartin A, Ezine E, Sassier M, et al. Cardiovascular immunotoxicities associated with immune checkpoint inhibitors: a safety meta-analysis n.d. (??) doi:10.1093/eurheartj/ehab708.

References:

1.     Tadic M, Cuspidi C, Hering D, Venneri L, Danylenko O. The influence of chemotherapy on the right ventricle: did we forget something? Clin Cardiol 2017; 40(7): 437-443. https://doi.org/10.1002/clc.22672.

2.     Sumin AN. Evaluating right ventricular function to reveal cancer therapy cardiotoxicity. Russian Open Medical Journal 2021; 10: e0309. DOI: 10.15275/rusomj.2021.0309

3.     Xu A, Yuan M, Zhan X, Zhao G, Mu G, Wang T, Hu H, Fu H. Early detection of immune checkpoint inhibitor-related subclinical cardiotoxicity: A pilot study by using speckle tracking imaging and three-dimensional echocardiography. Front Cardiovasc Med. 2022 Dec 21;9:1087287. doi: 10.3389/fcvm.2022.1087287.

Round 2

Reviewer 2 Report

I received answers from the authors to my questions and comments, they also made an amendment to the text of the manuscript. I have no other comments.